# A Scoping Review of Graphic Medicine Interventions to Promote Changes in Health Behavior, Health Service Engagement, and Health Outcomes

**DOI:** 10.3390/ijerph22050657

**Published:** 2025-04-22

**Authors:** Sarah Febres-Cordero, Athena D. F. Sherman, Biyeshi Kumsa, Meredith Klepper, Fawas Shanun, Sophie Grant, Brenice Duroseau, Sharon L. Leslie, Pranav Gupta, Abigail Béliveau, Patti Landerfelt, Sydney Cohen, Carissa Lawrence, Whitney Linsenmeyer, Molly Szczech, Monique S. Balthazar, Don Operario

**Affiliations:** 1Nell Hodgson Woodruff School of Nursing, Emory University, Atlanta, GA 30322, USA; adfsherman@emory.edu (A.D.F.S.); sophie.grant@emory.edu (S.G.); patti.e.landerfelt@emory.edu (P.L.); sydney.cohen3@emory.edu (S.C.); molly.szczech@emory.edu (M.S.); 2Rollins School of Public Health, Emory University, Atlanta, GA 30322, USA; biyeshikumsa9@gmail.com (B.K.); fawaz.shanun@emory.edu (F.S.); don.operario@emory.edu (D.O.); 3School of Nursing, Johns Hopkins University, Baltimore, MD 21218, USA; mkleppe1@jhmi.edu (M.K.); bdurose1@jhmi.edu (B.D.); clawre15@jhmi.edu (C.L.); 4Woodruff Health Sciences Center Library, Emory University, Atlanta, GA 30322, USA; sharon.leslie@emory.edu; 5School of Medicine, Emory University, Atlanta, GA 30322, USA; pranav.gupta@emory.edu; 6School of Nursing, University of North Carolina at Chapel Hill, Chapel Hill, NC 27599, USA; ajulier@ad.unc.edu; 7Nutrition and Dietetics, Saint. Louis University, Saint. Louis, MO 63103, USA; whitney.linsenmeyer@health.slu.edu; 8Ross and Carol Nese College of Nursing, Pennsylvania State University, University Park, PA 16802, USA; msb6243@psu.edu

**Keywords:** graphic medicine, art, comics, intervention design, intervention mediums, health literacy, equity

## Abstract

Low health literacy is a known contributing factor to poorer patient outcomes. Health information is often presented through materials written at high reading levels and thus may be an ineffective education tool for patients of diverse socioeconomic backgrounds, age ranges, and education levels. Graphic medicine (i.e., healthcare concepts presented through illustrations, such as comics or cartoons) may be a more equitable and efficacious format for many patients. The purpose of this review was to describe the efficacy and use of graphic medicine interventions regarding health outcomes, behavior changes, and engagement with health services. Nine databases were searched for studies that were randomized controlled trials in the English language, published before 4 December 2023. The searches identified 34 research articles that met the inclusion/exclusion criteria. This review revealed four key takeaways: (1) graphic medicine interventions are used globally; (2) graphic medicine interventions may be efficacious for a wide variety of health topics; (3) graphic medicine can be equitably delivered in many formats; and (4) graphic medicine can be applied broadly across the lifespan. The findings suggest that graphic medicine enhances patient engagement, empowers individuals with knowledge, and ultimately contributes to improved health outcomes across various populations; however, more effectiveness trials are needed. Additionally, an expanded definition of graphic medicine is presented.

## 1. Introduction

Health literacy is defined by the Centers for Disease Control and Prevention (CDC) as “the degree to which individuals have the ability to find, understand, and use information and services to inform health-related decisions and actions for themselves and others” [1]. With over one-third of Americans having low health literacy [2,3] health information is often difficult for the general public to understand [4]. Stossel et al. (2012) [5] found that the majority of patient education materials available at the point of care were written at reading levels considerably higher than that of the average American adult. Therefore, health communicators need better ways to inform patients and communities about health topics. Graphic medicine, which Czerwiec defines as “the intersection of the medium of comics and the discourse of health care”, presents opportunities to make information about health clearer and more approachable [6]. Despite graphic medicine’s potential to improve health education, little research has investigated its effectiveness in improving health outcomes and behaviors.

### What Is Graphic Medicine?

Graphic medicine tells stories using a combination of text and visuals, often employing comic components such as panels, images, speech, and thought balloons [7]. The National Library of Medicine defines graphic medicine as “the use of comics to tell stories of illness and health” [8]. By pairing illustrations with text, graphic medicine can provide an approachable medium for presenting complex health experiences that can be used by peers and healthcare providers to promote effective understanding. A seminal systematic review of the literature on the use of illustrations combined with text in health messaging found that such techniques improve attention to, and the recall of, health information, facilitate comprehension of the message, and influence intentions to act [9]. Given its benefits, graphic medicine is applicable across a variety of sectors, including public health, medicine, and education [10]. For example, graphic medicine can be used in medical education to boost empathy and improve communication skills among resident physicians [11,12]. In a qualitative survey of resident physicians participating in a novel curriculum featuring graphic medicine, 97% of residents reported the graphic medicine sessions to be a good use of their time [12]. In addition, graphic medicine can be helpful for patient education in hospital settings, particularly when explaining unfamiliar or anxiety-inducing concepts like surgery [13,14].

Although comics are often thought of as a medium for children, graphic medicine may be beneficial for both children and adults. For children, the benefit of illustrations combined with text in health education is well studied: one randomized study of 364 children found that story-format materials significantly improved comprehension compared to standard text-format materials [13]. In a more recent systematic review, graphic methods for achieving pediatric outcomes were explored, and pictorial images were found to be a preferred method when working with children who struggle with reading and text-based reporting measures [14]. In adults, health information presented with comics has also proven to be more informative than standard materials [14]. In a randomized trial comparing the efficacy of two vaccine information flyers, the comic flyer had a statistically significant effect on participants’ attitudes and perceptions of the flyer’s informativeness compared to the control flyer developed by the Centers for Disease Control and Prevention (CDC) [14].

Graphic medicine can be especially useful for providing health education for disenfranchised populations. The levels of health literacy are lower across many marginalized groups, including minoritized people, people without a high school degree, and people living in poverty [3,15]. Moreover, people with low health literacy are more likely to have poor health [16]. By increasing the accessibility of health information, graphic medicine can benefit diverse and marginalized populations [17]. Graphic medicine can also play a role in destigmatizing certain topics that may be difficult for people to discuss. For instance, a growing body of comics explores areas such as mental health [18], complex trauma [19], overdose response [20], death and dying [21], and HIV/AIDS [22].

Despite the ability of graphic medicine to improve understanding and reduce stigma, little research has investigated the effectiveness of these interventions in improving health. Thus, we conducted a scoping review to answer the following questions:

What is graphic medicine?What are the modes of delivery often used in graphic medicine interventions?What population groups have graphic medicine interventions been designed with and for?What are the efficacy and/or effectiveness of graphic medicine interventions regarding health outcomes, behavior changes, or improved engagement with health services/care?

## 2. Materials and Methods

The search strategy, study selection, and data extraction were organized and are reported in accordance with the PRISMA extension for scoping reviews (PRISMA-ScR [23]), and this review is registered with the PROSPERO international prospective register of systematic reviews (CRD42022384477).

### 2.1. Inclusion/Exclusion Criteria

This review included experimental studies with a comparison or control group published in peer-reviewed articles, in the English language, and on humans. Studies were sought that reported on the efficacy or effectiveness of a graphic medicine intervention regarding behavioral changes (health self-management behaviors, coping behaviors, preventative behaviors, etc.); health outcomes (physical or mental); and health service engagement (accessing care, seeking care, obtaining primary care, attending treatment, continued engagement in treatment). Graphic medicine was defined as still imagery (not videos or video games) that promotes health education. No limit was put on the publication date. Records were excluded if they were gray literature, dissertations, or if the graphic medicine intervention was part of a larger multicomponent intervention where the effects could not be parsed out.

### 2.2. Literature Search Strategy

A comprehensive literature search strategy was developed and conducted by an experienced medical librarian with input from the research team to identify relevant articles. Search terms were derived from a previously published scoping review on graphic medicine [24]. The searches combined controlled vocabulary supplemented with keywords related to the concept of graphic medicine (e.g., cartoon, comic book) and trial design (e.g., randomized, control). The draft strategies were peer-reviewed by another medical librarian and retested.

Searches were initially performed on 12 December 2022 and re-run for updates on 4 December 2023. Nine bibliographic databases were searched: APA PsycInfo (EBSCOhost); CINAHL (EBSCOhost); the Cochrane Database of Systematic Reviews; Embase.com; ERIC (EBSCOhost); Health Source: Nursing/Academic (EBSCOhost); LGBTQ+ Source (EBSCOhost); PubMed; and Scopus (Elsevier). The search strategy for PubMed may be found in Figure 1. Full search strategies for each database may be found in Appendix A.

### 2.3. Study Selection

A total of 3577 citations from the database searches were uploaded to EndNote [25], which identified and excluded 1413 duplicates. This left 2164 studies, which were uploaded to the Covidence systematic review software platform [26]. Covidence identified 195 additional duplicates, and the study team identified 5 additional duplicates, leaving 1964 records. Title and abstract screening for eligibility was performed by two independent investigators according to the inclusion/exclusion criteria. Conflicts between the reviewers were resolved by a third reviewer. Of these records, 1803 were excluded for irrelevancy, leaving 161 eligible for a full-text review. Of these, 127 were excluded, leaving 34 for data extraction and synthesis. The review and selection processes for the studies are summarized in the diagram in Figure 2.

Additionally, the following websites were hand-searched: https://www.graphicmedicine.org/book-series/graphic-medicine-manifesto/ (accessed on 12 December 2022 and again for updates on 4 December 2023), https://www.nlm.nih.gov/exhibition/graphicmedicine/index.html, https://repository.escholarship.umassmed.edu/ (accessed on 12 December 2022 and again for updates on 4 December 2023), and https://www.psupress.org/books/series/book_SeriesGM.html (accessed on 12 December 2022 and again for updates on 4 December 2023), resulting in 0 additional articles for review.

### 2.4. Article Data Extraction and Synthesis

Data were extracted by two review authors and audited by a third author. Matrix tables were developed in Excel and Microsoft Word to extract data, which were then categorized by our two-member results team by our a priori outcomes (behavior changes, health, and health services used), populations, and interventions, as recommended by Maxwell (1996) [27]. As these categories were developed, we began a deductive and inductive analysis using an iterative process to describe and interpret our results via descriptive content analysis [28].

## 3. Results

To assess and compare the various graphic medicine articles included in this review, we extracted information on the author, year, purpose, research design, total number of participants, population of interest, outcomes of interest, timepoints, intervention and comparison groups, and key findings (see Table A1 in Section A.2). In line with the inclusion criteria, all 34 articles employed a randomized control trial (RCT) research design, with some classified more explicitly as quasi-experimental (*n* = 2), cluster randomized controlled trials (*n* = 1), or non-inferiority controlled trials (*n* = 1). As intended, all articles included in this scoping review reported on the efficacy or effectiveness of a graphic medicine intervention regarding either behavioral changes, health outcomes, or health service engagement. The following results are presented to address our research questions: What is graphic medicine? What are the modes of delivery often used in graphic medicine interventions? What population groups have graphic medicine interventions been designed with and for? What are the efficacy and/or the effectiveness of graphic medicine interventions regarding health outcomes, behavior changes, or improved engagement with health services/care?

### 3.1. Defining Graphic Medicine

We found that in the context of healthcare, graphic medicine is a versatile medium that is not easily defined. From the use of graphic elements to sequential comics, graphic medicine was exemplified in multiple forms (Table A1). Traditional narrative and sequential comics were used throughout the studies to convey health-related information, engage readers, and facilitate learning. Authors often referred to comics using different descriptors: comics [29,30], comic leaflets [31,32], educational comics [33,34], comic books [35,36,37,38], cartoon pictures [39], cartoons [40,41], brochures with cartoon images [42], and manga [43,44,45] (Figure 3). Products that were non-sequential or lacked narrative linking using graphics included the following: illustrated books [46,47,48], printed materials [49], pictographs [50], children’s books/brochures [51], education books [52], pictorial aids [53], stories and coloring activities [54], pictorial images [55], graphic narratives [56], and educational cartoons [57] (Figure 4).

As a result of these findings, our augmented definition of graphic medicine is as follows:

Graphic medicine is a versatile medium that employs visual storytelling techniques, such as comics, graphic novels, illustrations, and cartoons, to convey medical information, educate patients about their conditions, improve treatment adherence, and facilitate communication between patients and healthcare providers. It encompasses a wide range of formats, including traditional narrative comics, manga-style illustrations and formats (read from the traditional back to the front), and graphic aids, each uniquely suited to addressing various healthcare needs and behavior change goals across different stages of life, from youth to end-of-life care. Graphic medicine interventions are designed to be engaging, understandable, and culturally relevant, potentially reaching diverse populations and promoting health equity by overcoming barriers such as low health literacy, language barriers, and disabilities affecting verbal communication. Through colorful illustrations, relatable characters, and age-appropriate language, graphic medicine interventions aim to alleviate anxiety and distress, enhance the understanding of medical concepts, promote adherence to treatment regimens, and facilitate better communication between patients, caregivers, and healthcare providers.

### 3.2. Graphic Medicine Design Processes and Modes of Delivery

Most developers began with a thorough literature review to ensure evidence-based content and an understanding of the existing interventions. The amount of community or end user involvement in the design process varied among the included articles. However, many studies intentionally engaged numerous key informants—such as educators [36], students, healthcare professionals [47], end users, and end user support people (e.g., parents, spouses, etc. [14,39,41,51])—to ensure that the content and format effectively addressed the needs and preferences of the target audience. Several research teams used an iterative process, which included several step-wise examinations of the developing interventions to ensure the content was relevant and acceptable to end users [38] and for those with disabilities such as hearing impairment [46], and significant community engagement, considering the demographic characteristics and needs of the low-socioeconomic-status population [51]. A simplistic intervention design was found in Kripalani et al. (2012) [47], wherein clip art such as pictures of eggs and bacon in a calendar format was utilized, indicating minimal effort and thought in the design of the educational materials. Some studies had limited engagement with communities and/or end users and other key informants [13,40,58]. However, other studies demonstrated a strong commitment to community consultation [58], the incorporation of literature reviews [49], interdisciplinary collaboration [33], and cultural adaptations [40].

The modes of delivery varied widely, often depending on the material and target audience. For instance, several articles used digital delivery. Imamura et al. (2014) [43] opted for an online approach, offering participants a series of stress management lessons delivered through a manga comic format. Other articles described in-person delivery with accompanying one-to-one counseling/instruction. Garcia de Avila et al. (2022) [49] paired an educational comic with a preoperative guidance session delivered by nurses, which included verbal information combined with the comic book. Some interventions also featured interactive components to enhance the learning and retention of health-related information [34,35,37,41]. While these interventions shared common characteristics, variations may have existed in the specific topics covered, format, and delivery methods used, influenced by cultural and intervention contexts and making a formal meta-analysis difficult [32,34,40,58].

### 3.3. The Application of Graphic Medicine Across the Lifespan

Graphic medicine can be used across the life course, and this was reflected in the included articles. We present the age distribution according to the stages of development according to Erikson as this theory focuses on development across the lifespan [59]. Seven of the eight stages were identified in this review (Figure 5).

#### 3.3.1. Youth

A graphic medicine approach provides a unique platform to address the healthcare needs of children and adolescents, presenting complex medical information in visually appealing and accessible formats. Interventions using graphic medicine for youth typically incorporate visually appealing elements such as colorful illustrations and relatable characters to capture the attention and interest of children and adolescents. Tailored to the age and developmental stage of the target audience, the content of these interventions addresses various medical topics, including those related to health conditions, such as safe sleep education, the prevention and management of low back pain, concussion recognition, the prevention and management of epilepsy, medical procedures, treatment regimens, and preventive care measures [33,34,35,36,51]. Emotional engagement is often incorporated to connect with youth on a deeper level, addressing feelings of anxiety [29,31,32,49,60], distress [43,52], or fear related to medical experiences, while portraying positive outcomes and coping strategies [32,45,48,54]. Numerous studies, including those by Kassai et al. (2016) [31], Kolberg et al. (2021) [33], and Nestadt et al. (2019) [41], have demonstrated graphic medicine’s efficacy in improving health outcomes among youth. These interventions have been shown to reduce anxiety and depression, enhance the understanding of medical concepts, promote adherence to treatment regimens, and facilitate better communication between patients, caregivers, and healthcare providers. Additionally, graphic medicine was used in Lusiana et al. (2023) [30] to influence behavior changes aimed at improving the nutritional knowledge and behavior of elementary school students.

#### 3.3.2. Adults

Having demonstrated its versatility and potential to improve health outcomes among youth, graphic medicine seamlessly transitions into adult healthcare, where its multi-faceted roles become increasingly evident. For instance, Brand et al. (2019) [29] explored its impact on patient comprehension and anxiety reduction before coronary procedures, suggesting its efficacy in alleviating anxiety and enhancing understanding among adults facing medical interventions. In a familial context, Bhana et al. (2004) [40] demonstrated how graphic medicine facilitates communication about sensitive topics within families, employing cartoons and narratives to ease discussions on issues like AIDS transmission and stigma among parents and pre-adolescent children. Similarly, Bazzano et al. (2023) [60] demonstrated how graphic medicine effectively addresses the needs of adults, particularly in the context of healthcare anxiety. By using a graphic novel as an intervention for patients awaiting oral biopsies, the study illustrated the power of visual storytelling in conveying complex medical information and alleviating anxiety. Through colorful vignettes depicting the biopsy procedure and its aftermath, the graphic novel provided patients with a visual narrative that enhanced their understanding of the process while also mitigating their fears and anxieties. This approach catered to adults by acknowledging their cognitive abilities and emotional experiences, offering a unique blend of information dissemination and emotional support that was accessible and engaging.

#### 3.3.3. End of Life

As individuals progress through the lifespan, graphic medicine continues to be relevant in end-of-life care. Studies like that by Ke et al. (2021) [42] emphasize its importance in facilitating communication and understanding between older patients and their surrogates regarding end-of-life care preferences. By providing a visual medium to explore themes of mortality and grief, graphic narratives help bridge the communication gap between patients and their families, ensuring that end-of-life care preferences are aligned. Evidence from these studies underscores the broad applicability of graphic medicine in addressing healthcare needs and behavior change goals across different stages of life, from youth to end-of-life care.

### 3.4. Application of Graphic Medicine Among Diverse Topics and Populations and Health Equity

In addition to its broad applicability across different stages of life, graphic medicine serves as a versatile tool for addressing a multitude of health-related issues among varying populations. Zhou et al. (2023) [61] and Shin et al. (2022) [38] demonstrated the application of graphic medicine among ethnically and racially diverse populations, specifically targeting Mexican American women and East African Americans, respectively. These studies placed heavy emphasis on using culturally relevant narrative messages to reduce health disparities associated with various behaviors, thereby promoting health equity. Graphic medicine is uniquely positioned to increase understanding and acceptability among those with low literacy levels, language barriers, or disabilities that may affect verbal communication [33,46,55]. By using such visuals, one can bridge communication gaps and ensure that important information is effectively conveyed to all end users [38,41,53,55,61], regardless of their individual characteristics or circumstances.

An intervention by Arunakul et al. (2012) [46] was specifically designed for hearing-impaired children aged 6–10, considering their communication needs and learning abilities. The use of sign language and clear illustrations in an illustrated book catered to the unique requirements of this population. Thus, using visual aids and sign language may have facilitated the better comprehension and retention of oral health information among participants. Moreover, sleep problems are common in children with learning disabilities, but effective interventions are scarce. Montgomery et al. (2004) [48] aimed to address sleep problems in learning-disabled children aged 2–8, which are often prevalent and challenging to treat. Additionally, the booklet was designed to be easy to understand, making it accessible to parents with varying literacy levels. This approach aimed to overcome barriers to accessing traditional face-to-face treatments, such as limited resources and professional expertise. By providing accessible and convenient interventions, this approach could potentially improve the quality of life for both children and their families.

### 3.5. Efficacy and Effectiveness of Graphic Medicine Interventions

A total of *n* = 34 graphic medicine studies were identified for this review and spanned countries across five of the six inhabited continents. These studies employed a variety of randomized controlled trial (RCT) designs, including randomized non-inferiority controlled trials [54], cluster randomized controlled trials [35], and quasi-experimental designs [30,42], among others. See Table A1 in Section A.2. This review illuminated mixed findings on the efficacy of graphic medicine and, although studies may have claimed to report on effectiveness trials, only one effectiveness study was identified, making it difficult to report on the effectiveness of graphic medicine as an intervention medium. The populations, methods, sample sizes, and control groups varied. Kulkarni et al. (2022) [32] sought to reduce preoperative anxiety in children aged 6–12 in an efficacy trial with a comic leaflet intervention with oral instructions compared to oral instructions alone (control). This study reported no significant reduction in preoperative anxiety compared to the control group. Conversely, in a proof-of-concept pilot study of preprocedural anxiety, Brand et al. (2019) [29] compared an informed consent standard (control) to informed consent with a comic supplement, finding a significant reduction in anxiety in the comic book group (*p* < 0.001). Similarly, infant sleep knowledge improved among mothers with a low socioeconomic status in the efficacy trial ‘Sleepy Baby: Safe and Snug’ [51]. Sleep knowledge was improved in both the ‘Sleepy Baby: Safe and Snug’ intervention and control (brochure) groups, while the intervention led to the behavior modifications of less bed sharing and exclusive crib use [51]. Kripalani et al. (2012) [47] aimed to test the efficacy of an illustrated medication schedule for medication adherence. No significant differences were found compared to controls. Simple graphics were added to calendars to enhance health literacy (e.g., bacon and eggs to remind someone to take a statin). Although adherence was not statistically improved in that study, adherence to antiretroviral therapy among adolescents in Thailand improved after a different, culturally tailored, cartoon-based comic intervention (11 sessions) was delivered over six and nine months compared to standard care [41].

Lastly, Al-Yateem et al. (2018) [54] conducted the only effectiveness study in our sample. The non-inferiority trial of a graphic medicine coloring book play and distraction intervention, “Adam Goes to Surgery”, versus preoperative medication (control group) incorporated narrative storytelling to reduce anxiety among children during perioperative day surgery. The graphic intervention was multilayered as a distraction tool and included a narrative, which introduced equipment, personnel, and procedures, combined with coloring. The book was also read to the children by their parents, allowing them time to answer questions. However, the differences were not significant enough to prove superiority (modified Yale Preoperative Assessment Scale (MyPAS) (*p* = 0.941), State-Trait Anxiety Inventory for Children (*p* = 0.708)).

## 4. Discussion

This review of graphic medicine interventions revealed four key takeaways: (1) graphic medicine interventions are used globally; (2) graphic medicine interventions may be efficacious and effective regarding a wide variety of health-related topics; (3) graphic medicine can be equitably delivered in many formats; and (4) graphic medicine can be applied broadly across the lifespan. Historically, graphic medicine was rooted in comic book culture, and this review captured some of this history with the inclusion of Gillies et al. (1990) [58], which used a volume of Streetwize UK, a comic published in the 1980s and 1990s which was co-created with community members at the intersection of art and health. Since that time, graphic medicine interventions have spread across the globe. Despite our review criteria only including articles published in English, the included articles represented RCTs conducted in five of the six inhabited continents, with broad applications for a range of health topics. More specifically, graphic medicine interventions have been shown to be efficacious and effective in areas such as mental health, behavioral dynamics, chronic conditions, health promotion, and sensitive topics such as HIV/AIDS [35,41,43,53,58,60].

The studies in this review illustrate how graphic medicine interventions have been employed to tackle various health concerns among diverse groups of people and communities. From reducing anxiety before coronary procedures to improving mental health literacy and medication adherence to changing lifestyle habits, graphic medicine interventions have proven instrumental in enhancing patient understanding, promoting behavioral change, and facilitating better health outcomes. The included articles highlight the potential of graphic medicine to address health disparities and promote health equity by using culturally relevant and visually appealing mediums to deliver health information. This breadth of applications underscores the significance of graphic medicine as a versatile tool in modern healthcare, capable of addressing diverse health issues and empowering individuals across different stages of life.

Although graphic medicine is often considered especially useful and particularly appealing to children and adolescents [62], these interventions can be applied across the lifespan [63]. Graphic medicine and visual storytelling can facilitate the delivery of complex medical information in a format that is accessible and more easily digestible than traditional formats [64,65]. In turn, this can help reinforce health education for children and adults. The use of comics in the delivery of health information can have a normalizing effect, increasing feelings of hopefulness and decreasing feelings of isolation among people experiencing physical or mental illness, as well as those living with marginalized social identities [66,67,68].

### 4.1. Strengths and Limitations

There are several strengths of this scoping review, including dual-reviewer systematic article identification, dual-reviewer data extraction, and the systematic auditing of the extracted data and data synthesis. There are also limitations of the current review, including potentially missed articles due to the exclusion of gray material and dissertations, single-reviewer article quality appraisal, and the narrow scope of particular health-related outcomes. Due to the broad nature of the inclusion criteria, which selected for (1) experimental, (2) human-centered (3) studies in the English language (4) with a comparison or control group, (5) published in peer-reviewed articles, this scoping review covers articles on a variety of graphic medicine interventions. All interventions, though different in form, met this review’s definition of graphic medicine: non-video, still imagery that promotes health education. However, despite meeting this criterion, many of the included graphic medicine interventions were fundamentally different. For instance, we recognize that manga and illustrated children’s books differ greatly in their target audience, level of sophistication, and overall structure. On the one hand, including a wide range of graphic medicine interventions may have made it difficult to draw conclusions about any one form of imagery or mode of delivery. On the other hand, this variation between articles is part of the scoping review process, and we intentionally employed broad inclusion criteria to capture a large number of peer-reviewed articles and to investigate the field of graphic medicine as a whole without favoring any one type of imagery or mode of delivery. Despite these limitations, this review provides a meaningful synthesis of the current state of the graphic medicine intervention RCT literature, which can be used to guide future intervention development and adaptation for diverse populations.

### 4.2. The Way Forward: Implications and Recommendations

Graphic medicine has been established as a novel source of diverse graphic narratives in varied contexts and should be broadly implemented across health disciplines. It has been nearly a decade since six pioneers of the field published the Graphic Medicine Manifesto [6]. This collection of essays stressed the need for diverse and interdisciplinary perspectives, inclusive of healthcare workers, patients, and educators. Although there are notable graphic medicine novels and related research articles across health disciplines, the formal integration of this topic has been fractured. Graphic medicine has notably been implemented in health curricula largely for medical students, particularly for boosting empathy and communication skills [12]. Graphic medicine could also make a significant impact in the nursing profession. As the largest segment of the healthcare workforce [69] and with a worsening nursing shortage in America [70] and globally [71], graphic medicine could be used as an innovative supplement to traditional health education. This may have positive effects on retention in training programs and may assist clinicians in delivering health content to patients, further supporting their retention in clinical care and improve patient knowledge, intention, and behaviors.

Recently, the Coronavirus Disease 2019 (COVID-19) pandemic represented a turning point in the scale of the use of graphic medicine to deliver health-related instructional, personal, and therapeutic content, largely via social media [72]. In 2020, graphic medicine became an important medium for alerting the public to the nature of the pandemic as well as for sharing pertinent information about transmissibility and safety. Graphics were created by the world’s most influential health organizations (e.g., CDC, WHO) and shared across social media platforms [72]. Consequently, globally, people of all ages, including older adults [63], vastly expanded their social media engagement. While the graphics and visual storytelling may have improved COVID-19-related health literacy, there were unintended consequences as the creation and dissemination of graphic medicine content became unregulated, and misinformation was shared by various sources [73]. This is a major concern for clinicians and public health officials moving forward, emphasizing the need for evidence-based and community-co-created graphic medicine and public education regarding how to identify a trusted source of health information.

This review highlights four key insights: (1) graphic medicine interventions are implemented worldwide; (2) they show potential efficacy across a broad range of health topics; (3) they can be delivered equitably through diverse formats; and (4) they are applicable across all stages of life. While the findings suggest that graphic medicine fosters patient engagement, supports knowledge empowerment, and may lead to improved health outcomes across diverse populations, more effectiveness trials are necessary to strengthen the evidence base.

## 5. Conclusions

With their highly accessible format, graphic medicine interventions show promising efficacy in improving health outcomes through patient engagement and knowledge empowerment. While further effectiveness trials are necessary to strengthen the evidence base, they have the potential to benefit diverse communities across a range of health topics throughout the lifespan through the use of culturally relevant materials in a variety of formats.

## Figures and Tables

**Figure 1 ijerph-22-00657-f001:**
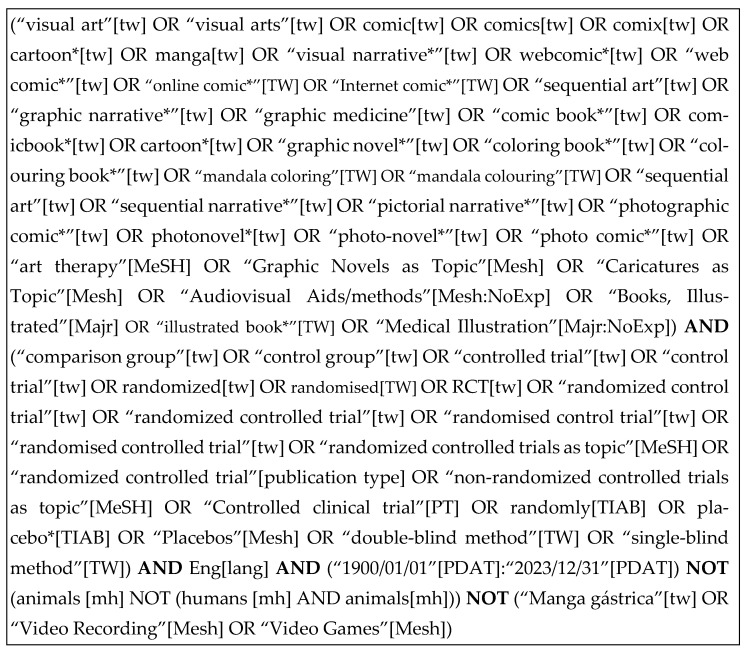
Search strategy for PubMed.

**Figure 2 ijerph-22-00657-f002:**
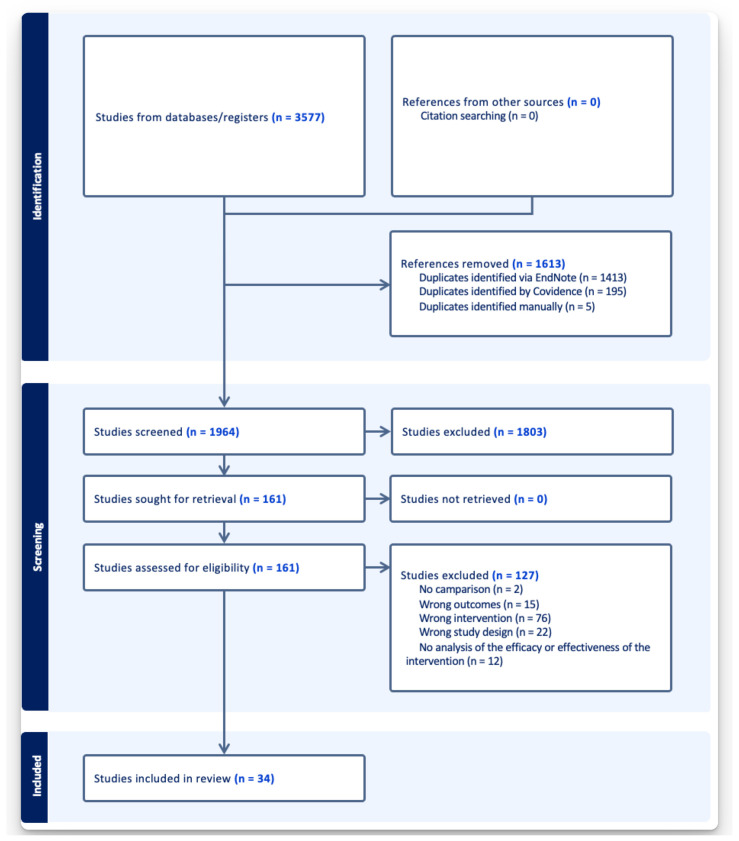
PRISMA 2020 flow diagram.

**Figure 3 ijerph-22-00657-f003:**
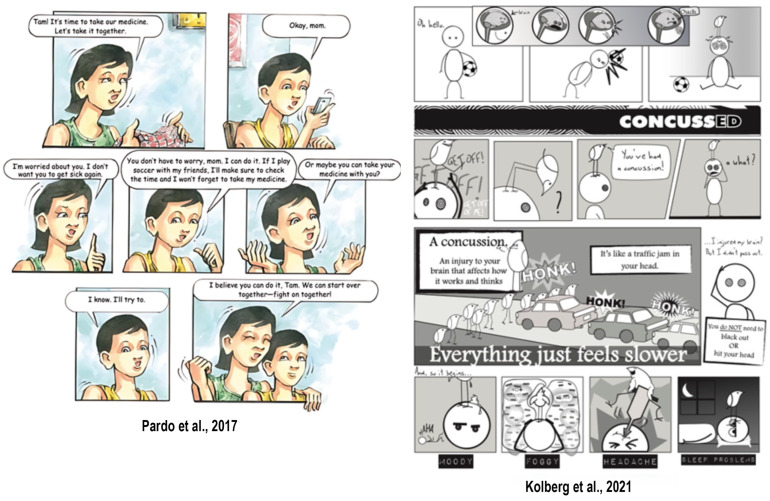
Example images of comic-style graphic medicine interventions [33,41].

**Figure 4 ijerph-22-00657-f004:**
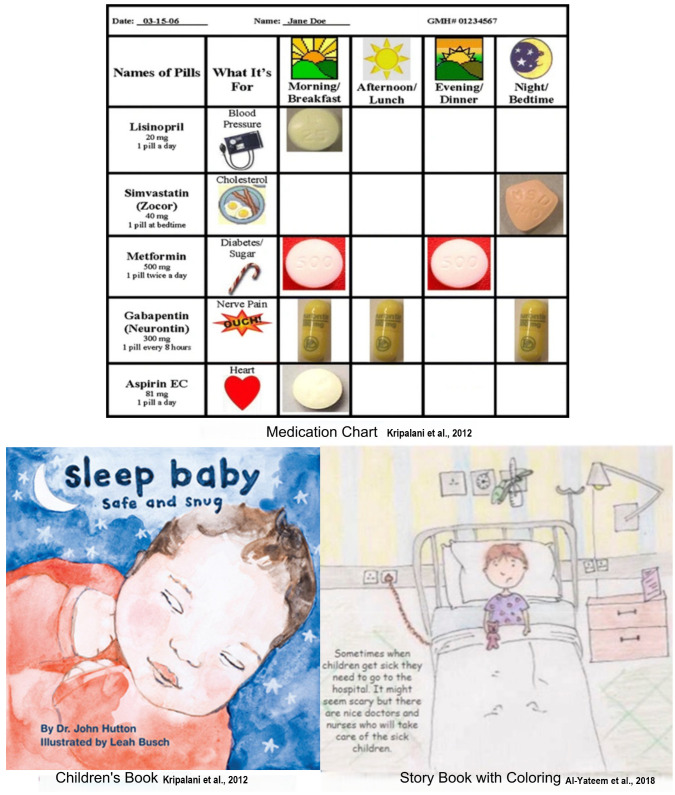
Example images of non-comic-style graphic medicine interventions [47,51,54].

**Figure 5 ijerph-22-00657-f005:**
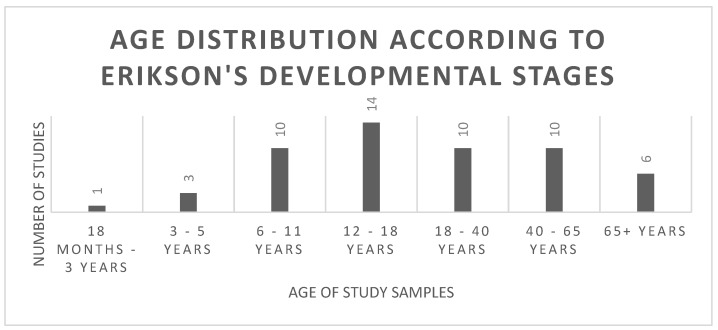
Age distribution according to Erikson’s Developmental Stages.

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
