# Peer review of "A Scoping Review of Graphic Medicine Interventions to Promote Changes in Health Behavior, Health Service Engagement, and Health Outcomes"

_ijerph, 2025, doi:10.3390/ijerph22050657_

Round 1

Reviewer 1 Report

Comments and Suggestions for Authors

Trying to go through the long long long multipage table which you have chosen as a way to summarize what is being said by publications you think useful is difficult at best. Of course you had to create the table - it encapsulates your findings. And it makes a wonderful appendix. So I wonder if you cannot summarize  several of the different columns, and I'd suggest Purpose, Research Design, Outcomes, and Key Findings. Perhaps you might consider dividing the summaries by ages and then by medical fields?  And you might consider expanding your discussion of genre:  anga and books for children are just not the same.  This is going to be a highly valuable study - I think it needs to be made more aware of what its readers might need.

Comments on the Quality of English Language

Your English is fine; you just have occasional typos outside of the table and I highlighted several.

Author Response

We would like to thank the reviewers of our work. Peer review is such an important part of the process of presenting our work. We worked to address all comments. Please see reviewer comments and response below.

With gratitude,

Sarah Febres-Cordero and Athena DF Sherman

 Reviewer 1

Comment:

Trying to go through the long long long multipage table which you have chosen as a way to summarize what is being said by publications you think useful is difficult at best. Of course you had to create the table - it encapsulates your findings. And it makes a wonderful appendix.

Response: Thank you for pointing this out. We agree with your comment. Therefore, we have moved Table 1. To the appendix A.2.1, line #518

Comment: I wonder if you cannot summarize  several of the different columns, and I'd suggest Purpose, Research Design, Outcomes, and Key Findings. Perhaps you might consider dividing the summaries by ages and then by medical fields? 

Response: Thank you for this thoughtful critique. To better present the results of the review we have added a paragraph page 6 lines 195-208 to introduce the findings as related to the research questions. We think this addition clarifies our summaries.

Comment: You might consider expanding your discussion of genre:  Manga and books for children are just not the same. 

Response: Thank you for pointing this out, and we agree, they are not the same. We have addressed this point the limitations section of the manuscript. Page 11 lines 440-456.

Reviewer 2 Report

Comments and Suggestions for Authors

Overall, I found this article to be well-written and well-done!  My area of interest is health literacy and I can definitely see the value of graphic medicine interventions for both patients and caregivers across the lifespan.  

There are a few paragraphs that start with The The (under Materials and Methods), In In (120) and This This (146).  Note that line numbering doesn't start at the beginning on the copy I printed.

Line 60, has a Figure 5, Age Distribution According to Erikson's Developmental Stages, but Line 65 labels this as Figure 2 (error?)  Another line or two about this Figure might be helpful (perhaps according to Erikson's Stages).

Author Response

We would like to thank the reviewers of our work. Peer review is such an important part of the process of presenting our work. We worked to address all comments. Please see reviewer’s comments and response below.

With gratitude,

Sarah Febres-Cordero and Athena DF Sherman

Reviewer 2:

Comment: There are a few paragraphs that start with The The (under Materials and Methods), In In (120) and This This (146).  Note that line numbering doesn't start at the beginning on the copy I printed.

Response: Thank you for catching these errors. We had the entire manuscript edited and all errors have been addressed throughout the manuscript.

Comment: Line 60, has a Figure 5, Age Distribution According to Erikson's Developmental Stages, but Line 65 labels this as Figure 2 (error?)  Another line or two about this Figure might be helpful (perhaps according to Erikson's Stages).

Response: Thank you for pointing out the error in the figure numbering. We have also added sentences to contextualize the use of Erikson for the figure. Lines 273-275

Reviewer 3 Report

Comments and Suggestions for Authors

A lot of work must have gone into this scoping review and I would first want to express my respect for the way the research was conducted, respecting all the norms, and for your painstaking analysis of the results.

That said, I think the findings are relatively limited in value. You expand the definition of graphic medicine and show that interventions can be used widely, addressing various age groups and social groups. The main merit of your article is that it heightens awareness of the role and potential of graphic medicine and that it points out that more research is needed to gain evidence of its effectiveness.

Table One shows how fine-grained your analysis is, but it interrupts the flow of the argument and had better be presented separately. This would considerably improve the readability of your article.

I am attaching more details in a separate file.

Author Response

We would like to thank the reviewers of our work. Peer review is such an important part of the process of presenting our work. We worked to address all comments. Please see reviewer’s comments and response below.

With gratitude,

Sarah Febres-Cordero and Athena DF Sherman

Reviewer 3:

Comment: As indicated above, the whole of Table 1 had better be made available separately (and in a legible format).

Response: Thank you for pointing this out. We agree with your comment. Therefore, we have moved Table 1. To the appendix A.2.1, line #518.  We have also changed the layout to landscape to make the table legible.

Comment: Furthermore, the text of the article needs careful proofreading. Some examples of imperfections:

Response: Thank you for catching these errors. We had the entire manuscript edited and all errors have been addressed throughout the manuscript.

Comment: Abstract: “in the English” → _in the English language Top p3: missing full stop Repetitions of words: Top p. 4 (The The), line 1 on p 41 (Kulkarni Kulkarni), line 120 (In In), line 140 (This This) 2 pages

Response: Thank you for catching these errors. We had the entire manuscript edited and all errors have been addressed throughout the manuscript.

Round 2

Reviewer 1 Report

Comments and Suggestions for Authors

Much better. Now you need to add something to p 13 to conclude your brief discussion since everything else seems to be your tables. Look at your abstract and add a slightly revised version of "This review revealed four key takeaways: 1) graphic medicine interventions are used globally 2) graphic medicine interventions may be efficacious in a wide variety of health topics; 3) graphic medicine can be equitably delivered in many formats; and 4) graphic medicine can be applied broadly across the lifespan. Findings suggest that graphic medicine enhances patient engagement, empowers individuals with knowledge, and ultimately contributes to improved health outcomes across various populations, however, more effectiveness trials are needed."

Author Response

Reviewer 1

Comment: Much better. Now you need to add something to p 13 to conclude your brief discussion since everything else seems to be your tables. Look at your abstract and add a slightly revised version of "This review revealed four key takeaways: 1) graphic medicine interventions are used globally 2) graphic medicine interventions may be efficacious in a wide variety of health topics; 3) graphic medicine can be equitably delivered in many formats; and 4) graphic medicine can be applied broadly across the lifespan. Findings suggest that graphic medicine enhances patient engagement, empowers individuals with knowledge, and ultimately contributes to improved health outcomes across various populations, however, more effectiveness trials are needed."

Response: Thank you so much for taking the time to review our manuscript. Your suggestions improved this work.

We added the revision of the lines in the abstract to the end of the discussion. Lines 510-516.

This review highlights four key insights: 1) graphic medicine interventions are implemented worldwide; 2) they show potential efficacy across a broad range of health topics; 3) they can be delivered equitably through diverse formats; and 4) they are applicable across all stages of life. While findings suggest that graphic medicine fosters patient engagement, supports knowledge empowerment, and may lead to improved health outcomes across diverse populations, more effectiveness trials are necessary to strengthen the evidence base.

We also took this round of revisions as an opportunity to edit the entire manuscript one last time. Sharon Leslie and Patti Landerfelt, co-authors, went through and added revisions throughout.

Reviewer 3 Report

Comments and Suggestions for Authors

The revised version addresses my earlier concerns and the article in its present form is now fit for publication. I have noticed that additional lines have been added in the section on 'limitations' (possibly as a reaction to comments from a second reviewer) and I am agreed with this addition.

By transferring the lengthy table to an appendix, the readability of the article has considerably increased.

Author Response

Comments: The revised version addresses my earlier concerns and the article in its present form is now fit for publication. I have noticed that additional lines have been added in the section on 'limitations' (possibly as a reaction to comments from a second reviewer) and I am agreed with this addition.

By transferring the lengthy table to an appendix, the readability of the article has considerably increased.

Response:

Thank you so much for taking the time to review our manuscript. Your suggestions improved this work.